

# Genome-wide identification of hypoxia-induced enhancer regions

Nick Kamps-Hughes, Jessica L. Preston, Melissa A. Randel and Eric A. Johnson

Institute of Molecular Biology, University of Oregon, Eugene OR, United States

## ABSTRACT

Here we present a genome-wide method for *de novo* identification of enhancer regions. This approach enables massively parallel empirical investigation of DNA sequences that mediate transcriptional activation and provides a platform for discovery of regulatory modules capable of driving context-specific gene expression. The method links fragmented genomic DNA to the transcription of randomer molecule identifiers and measures the functional enhancer activity of the library by massively parallel sequencing. We transfected a *Drosophila melanogaster* library into S2 cells in normoxia and hypoxia, and assayed 4,599,881 genomic DNA fragments in parallel. The locations of the enhancer regions strongly correlate with genes up-regulated after hypoxia and previously described enhancers. Novel enhancer regions were identified and integrated with RNAseq data and transcription factor motifs to describe the hypoxic response on a genome-wide basis as a complex regulatory network involving multiple stress-response pathways. This work provides a novel method for high-throughput assay of enhancer activity and the genome-scale identification of 31 hypoxia-activated enhancers in *Drosophila*.

## INTRODUCTION

Gene expression is differently regulated in different cell types and in response to changes to environmental conditions. This regulation is achieved in part by the activity of transcriptional enhancers (*Bulger & Groudine, 2011*; *Perry, Boettiger & Levine, 2011*; *Lagha, Bothma & Levine, 2012*; *Arnosti & Kulkarni, 2005*; *Swanson, Schwimmer & Barolo, 2011*), specific DNA sequences that bind transcription factors to control the rate of transcription initiated at nearby promoters. Even for relatively simple processes, such as the acute response to changes in oxygen availability, the identification and characterization of the enhancers used to shift the network of gene expression to a new mode remains limited.

The transcription factor hypoxia-inducible factor-1 (HIF-1) is directly inhibited by the presence of cellular oxygen via protein degradation of the HIF-1$\alpha$ subunit (*Bruick & McKnight, 2001*). Once stabilized, HIF-1$\alpha$ moves to the nucleus and up-regulates the transcription of target genes. Although HIF-1 remains a central regulator in models of how cells respond after experiencing low oxygen (*Lavista-Llanos et al., 2002*; *Wang & Semenza, 1993*), more recently other transcription factors have been implicated in the hypoxic

Corresponding author
Nick Kamps-Hughes,
nkampshughes@gmail.com

response in a complex network of regulatory events. For example, the immunity response transcription factor NF-KB is also activated by hypoxia and regulates the transcription of HIF-1 (*Rius et al., 2008*; *Van Uden et al., 2011*), while HIF-1 appears to play a reciprocal role in the regulation of NF-kB targets (*Scortegagna et al., 2008*). Likewise, HIF-1 sensitizes the heat shock response by directly regulating heat shock factor (HSF) transcription during hypoxia. Thus, the broader picture that has emerged is that the stress response transcription factor pathways are not isolated regulatory units but rather cooperate and co-opt each other to modify the cell's functions in a complex manner.

High-throughput sequencing tools have become widespread in gene expression studies (*Metzker, 2010*; *Wang, Gerstein & Snyder, 2009*; *Johnson et al., 2007*). For example, RNAseq has become a powerful tool for analyzing differential gene expression by quantifying the RNA abundance of the transcriptome. However, RNAseq does not provide empirical information about the regulatory events leading to a change in transcript abundance. ChIPseq provides information about where transcription factors bind to the genome, but binding events do not always result in an active enhancer or change in the rate of transcription. Another sequencing strategy assays open chromatin conformations (*Song et al., 2011*) as a reliable proxy for enhancers. However, until recently the typical functional assay for enhancers was to clone the putative regulator upstream of a reporter gene driven by a minimal promoter.

Several next-generation sequencing-based methods have been used to dissect the function of individual nucleotides within previously known enhancers (*Kwasnieski et al., 2012*; *Patwardhan et al., 2012*; *Melnikov et al., 2012*; *Kheradpour et al., 2013*) as well as scan genomic sequence for enhancer activity (*Arnold et al., 2013*). These methods have either used UTR tags to assay from thousands to hundreds of thousands of fragments in parallel (*Kwasnieski et al., 2012*; *Patwardhan et al., 2012*; *Melnikov et al., 2012*; *Kheradpour et al., 2013*) or have had to confine the potential enhancer itself to the UTR in order to assay genome-scale complexities (*Arnold et al., 2013*). Here we use a novel variation on these high-throughput enhancer screening methods to identify regions of the *Drosophila* genome with increased activity under hypoxia. Our technique combines randomly sheared genomic fragments to be assayed for activity with a UTR randomer tag system for highly multiplexed tracking of transcriptional activity. The construct library is modularly synthesized *in vitro* making the relative placement of construct elements easily mutable. This is in contrast to a similar method called STARR-Seq (*Arnold et al., 2013*) that requires the potential enhancer itself be placed downstream of the transcription start site. Although enhancers are known to function at variable distance and orientation with respect to a target promoter (*Bulger & Groudine, 2011*; *Banerji, Rusconi & Schaffner, 1981*) their strength has been shown to be modulated by their position relative to the target promoter (*Amit et al., 2011*) and transcriptional read-through has been shown to attenuate their activity (*Erokhin et al., 2013*). The method in this paper allows the regulatory element to be placed at the discretion of the experimenter. Additionally, the previously published library construction methods (*Kwasnieski et al., 2012*; *Patwardhan et al., 2012*; *Melnikov et al., 2012*; *Kheradpour et al., 2013*; *Arnold et al., 2013*) require microbial propagation of DNA libraries whereas we present a simpler entirely *in vitro* strategy. The work presented

here is the first implementation of a massively parallel reporter assay to study cis-regulatory activity during an environmental stress response. A library of 4,599,881 random 400–500 bp fragments spanning the *Drosophila melanogaster* genome was used to identify 31 hypoxic enhancer regions. The regions coincide with genes up-regulated under hypoxia and with binding site motifs from multiple transcription factors involved in the hypoxic response. This work provides mechanistic details of the hypoxic response by empirically identifying regulatory regions that drive hypoxic transcription, linking them to target genes from RNAseq differential expression data, and identifying trans-acting factors in silico. Investigating the hypoxic response in *Drosophila* allows us to corroborate previous work on hypoxic gene regulation (*Lavista-Llanos et al., 2002*; *Rius et al., 2008*; *Van Uden et al., 2011*) and a previous empirical genome-wide enhancer screen (*Arnold et al., 2013*). This genome-wide scan demonstrates the complexity of the hypoxic response, which involves multiple regulators acting in concert to control the expression of a wide variety of targets.

## MATERIALS AND METHODS

All DNA sequencing was performed on the Illumina HiSeq. All PCR reactions contained a final concentration of 400 nM of each primer and used Phusion Polymerase in 1X HF buffer. All oligonucleotide sequences are listed in File S1.

### Library synthesis

The linear reporter library used to assay enhancer activity was constructed entirely *in vitro* (Fig. 1A). The sequence space being assayed for enhancer activity, in this case the *Drosophila melanogaster* genome, was sonically sheared to generate random enhancer-sized fragments. Adapter ligation and 5′ PCR addition were used to add the Illumina first-end sequence upstream of the sheared DNA and part of the minimal promoter downstream. 5′ PCR additions are used to add minimal promoter elements, an intron to stabilize mRNAs (*Zieler & Huynh, 2002*), the 20 N randomer tag, and Illumina paired-end sequence upstream of an arbitrary ORF, in this case GFP. The synthetic minimal promoter used was designed to contain several core motifs and has been shown to function with a wide range of enhancers (*Pfeiffer et al., 2008*). The two fragments are then ligated together to create the final construct library pictured in Fig. 1A. The reporter library was diluted to a target of 10,000,000 molecules and regenerated by PCR so that the library could be adequately characterized by paired-end sequencing. An aliquot of the reporter library is used for paired-end sequencing to match randomer tags located in the 5′ UTR to the non-transcribed genomic region driving their expression. The library is then transfected into cells for massively parallel enhancer assay (Fig. 1B).

*Drosophila melanogaster* strain Oregon-R genomic DNA was sonically sheared using the BioRuptor. 400–500 bp fragments were isolated by gel electrophoresis then end-repaired using Blunt Enzyme mix (NEB) and 3′ adenylated using Klenow exo- (NEB). This sample was then ligated to an asymmetric adapter with T-overhang composed of annealed oligonucleotides Genomic-Adapter-1 and Genomic-Adapter-2. The ligation product was gel-purified and used as PCR template with primers Illumina P5 and Genomic-R to create a library of molecules containing a random 400–500 bp stretch of *Drosophila melanogaster*

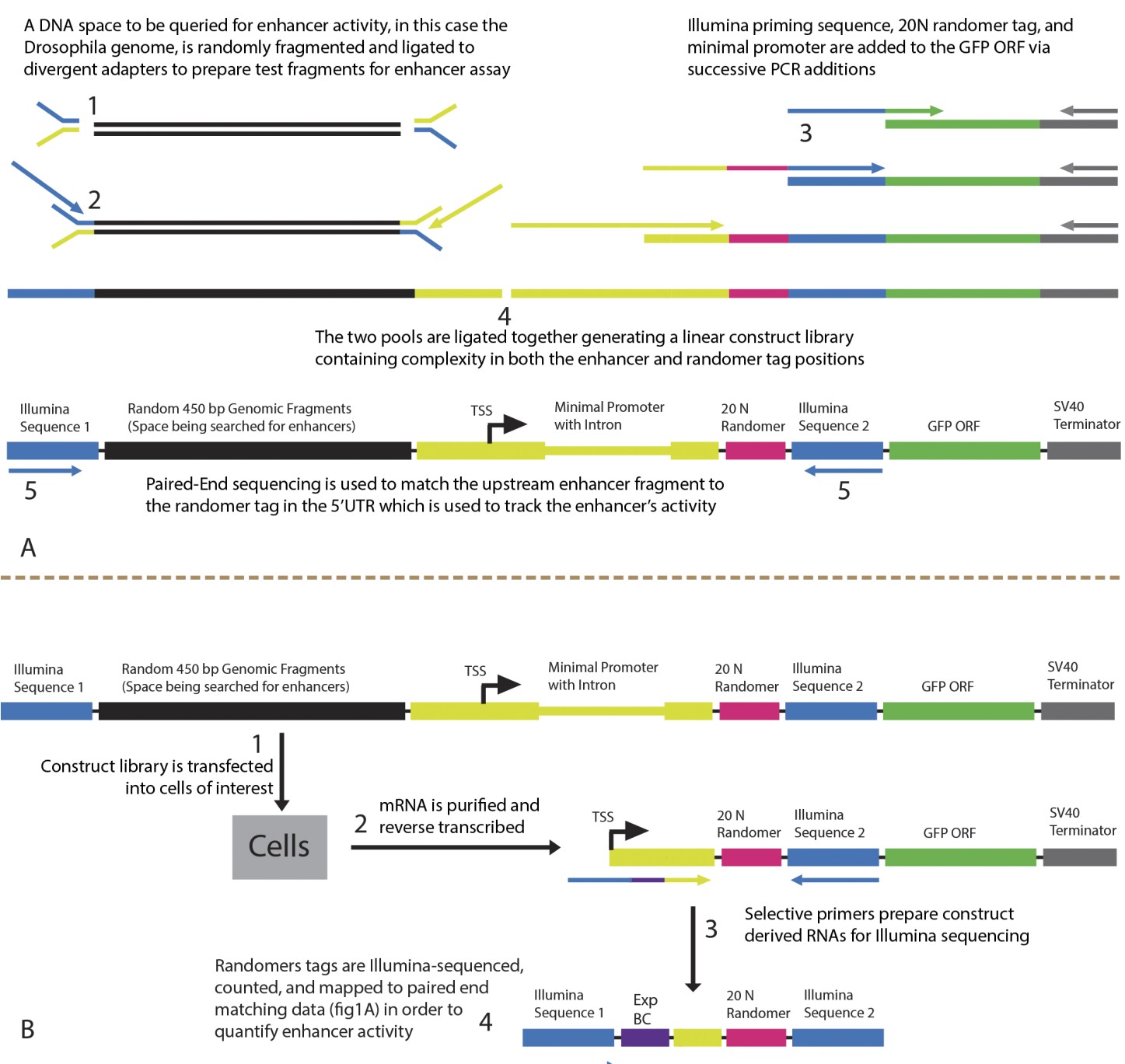

**Figure 1** **Library synthesis.** (A) The enhancer library is synthesized entirely *in vitro*. DNA of interest is fragmented (step 1) and ligated to divergent adapters (step 2) leaving potential enhancer fragments with Illumina sequence on one side and the beginning of the synthetic minimal promoter on the other. The GFP gene is used as a template for a series of 5′ PCR additions in order to add Illumina sequence, 20 N randomer tag, and the majority of the minimal promoter and intron (step 3). The two sides are ligated together to create a linear construct with complexity in the enhancer region upstream of the transcription start site as well as complexity in the randomer tag region in the 5′ UTR (step 4). The sample is submitted to paired-end sequencing in order to match the potential enhancer region to the randomer tag 
genomic sequence between the Illumina end one sequence and the beginning of a synthetic promoter. Separately, The GFP coding sequence followed by the SV40 terminator was PCR amplified from plasmid pGreen-H-Pelican with primers GFP-F and SV40-R. This product was then used as template for a PCR reaction using primers SV40-R and Marker-1-F. This product was then used as template for a PCR reaction using primers SV40-R and Marker-2-F. This product was then used as template for a PCR reaction using primers SV40-R and Marker-3-F to create a library of molecules containing a GFP sequence downstream of a minimal promoter with randomer tag and Illumina paired-end sequences. The genomic sequence-containing library and minimal promoter library were then 3′ adenylated and 3′ thymidylated respectively with Klenow exo- then ligated together. The heterodimer (1,819–1,919 bp) was gel-purified and subsequently selected for proper orientation by PCR with primers SV40-R and Illumina P5. To reduce library complexity to a scale that was tractable by paired-end sequencing, DNA was quantified using the Qubit system (Invitrogen) and serially diluted to produce an estimated 10,000,000 molecules that were used as template to regenerate the library by PCR with primers SV40-R and Illumina P5. An aliquot of this library was used as template for a PCR reaction with primers Illumina-P7 and Illumina-P5 to generate a paired-end Illumina-sequencing library such that the first-end sequence contained the beginning of the genomic region and the paired-end sequence contained the corresponding randomer tag (Fig. 1A). Aliquots were also used to generate transfectable quantities of the full-length reporter library by PCR amplification of the entire fragment using primers SV40-R and Illumina-P5. The final construct library sequence is available in File S2.

## Transfection, RNA extraction, and randomer tag sequencing

Six 5 mL flasks were plated to 80% confluency with S2 cells (contributed by Ken Prehoda lab, University of Oregon) and transfected with Fugene HD and 2.6 ug reporter library DNA at a 3:1 ratio. The following day three plates were placed under hypoxia (99.5% N2 and 0.5% O2) for five hours and thirty minutes and three were left in atmospheric conditions. Total RNA from both conditions was extracted using Trizol and treated with DNAse Turbo (Ambion, Foster City, CA, USA). RNA was converted to cDNA with SuperScipt III first strand synthesis kit (Invitrogen, Waltham, MA, USA) using oligo dT20 primers. cDNA was used as template for PCR with primers flanking the randomer tag to create an amplicon ready for Illumina sequencing. All PCR reactions used Illumina-P7 reverse primer and the following barcoded forward primers to allow multiplexing: RNA-BC-1 for hypoxic sample 1, RNA-BC-2 for hypoxic sample 2, RNA-BC-3 for hypoxic sample 3, RNA-BC-4 for normoxic sample 1, RNA-BC-5 for normoxic sample 5, RNA-BC-6 for normoxic sample 6. The resulting 178 bp amplicons were combined and sequenced on the Illumina Hiseq.

## RNAseq

RNA from the same experiments used to quantify enhancer activity was used for RNAseq. mRNA was purified using Dynabeads (Invitrogen, Waltham, MA, USA) from 10 ug of total RNA and chemically fragmented using Ambion Fragmentation Reagent. cDNA libraries were made with SuperScipt III first strand synthesis kit using random hexamer primers followed by second-strand synthesis with DNA Pol I (New England Biolabs, Ipswich, MA, USA). The double stranded DNA was end-repaired using NEB Quick Blunting Kit and 3′ adenylated using Klenow exo-. The samples were ligated to divergent Illumina adapters with in-line barcodes (Hypoxic GGTTC, Normoxic CTTCC) and PCR amplified with Illumina primers. 300–450 bp fragments were gel-purified and sequenced on the Illumina HiSeq (hypoxic condition: Accession SRX467593, normoxic condition: Accession SRX467591). 6,855,528 reads from each sample were aligned to the *Drosophila melanogaster* transcriptome (Flybase, r5.22) using TopHat (*Trapnell, Lior & Salzberg, 2009*). The bam outputs were analyzed by cufflinks and the resulting transcripts.gtf files were compared using cuffdiff to identify differentially expressed genes (File S3). Some ncRNAs were also analyzed for differential expression. Since they are not present in the transcriptome build, RNAseq reads were aligned to each ncRNA using Bowtie2 (*Langmead & Salzberg, 2012*) and their expression level is reported by normalized number of aligned reads in each condition.

## Computational enhancer activity analysis pipeline

All scripts and a tutorial are available in File S4 Paired-end fastq files (Accession SRX468157) linking genomic regions in the first-end read to randomer tags in the paired-end read were parsed to a fasta file with the randomer tag as the sequence name and the genomic sequence as the sequence. This file containing 32,061,029 sequences was aligned to the *Drosophila melanogaster* genome (dm3) using Bowtie2 (*Langmead & Salzberg, 2012*). Reads were processed into a match-list linking randomer tags to the genomic coordinates of their corresponding test sequence.

Randomer tags from hypoxic and normoxic RNA amplicon sequencing were extracted from fastq files (Accessions SRX468694, SRX468097) and experimental replicates were separated by barcode. 18,261,667 randomer tags from hypoxic sample 1, 14,226,458 from hypoxic sample 2, 14,697,154 from hypoxic sample 3, 14,406,854 from normoxic sample 1, 14,988,132 from normoxic sample 2, and 11,516,478 from normoxic sample 3 were referenced to the paired-end match list to generate genome-wide enhancer activity tables by 100 bp bins. The genomic fragments ranged from 400–500 bp so the bin corresponding to the alignment as well as the four downstream bins were credited 1 count. In the cases where randomer tags linked to multiple genomic fragments, bins were credited a fraction of a count based on the likelihood of that linkage in the paired-end match data. This created a count table of enhancer activity in each replicate at each 100 bp region of the *Drosophila melanogaster* genome.

The count table was then analyzed for differential activity between hypoxic and normoxic replicates using a negative binomial test in the DESeq (*Anders & Huber, 2010*) package within the R programming language. The bins were filtered by overall count ($\theta = 0.5$) and

the test was run with default variance estimation. This generated a $p$-value and a $p$-value adjusted for multiple hypothesis testing (Benjamini–Hochberg procedure) for each 100 bp bin. Hypoxic enhancer regions were defined at bins up-regulated under hypoxia with adjusted $p$-value $< 0.1$ ($p$-value $< 1.55e{-}05$) and extended to include adjacent bins with $p$-value $< 0.05$.

In order to compare our results to those of STARR-seq (*Arnold et al., 2013*), we also identified statistically significant S2 baseline enhancers within the normoxic replicates. In this case, the negative binomial test was performed between counts in the transfected DNA library and counts in the RNA-sequenced barcodes for each 100 bp bin across the *Drosophila melanogaster* genome in order to identify genomic regions enriched for baseline enhancer activity. Peaks were identified with an adjusted $p$-value $< 0.018$ (Benjamini–Hochberg procedure) and enhancers were defined as the 500 bp interval surrounding the activity peak in order to maintain consistency with STARR-seq data.

### Enhancer sequence motif analysis

Identified hypoxic enhancer regions were searched for stress transcription factor binding sites using the BoBro BBS motif-scanning algorithm (*Ma et al., 2014*) with position weight matrices from the JASPAR database (*Mathelier et al., 2013*). This algorithm was used to identify binding site positions and calculate a global $p$-value of enrichment for HIF-1 (JASPAR ID:MA0259.1), FOXO (MA0480.1), HSF (MA0486.1) and NF-kB (MA0105.3) binding sites in enhancer sequences compared to the *Drosophila melanogaster* genome background.

## RESULTS

### Discovered hypoxic enhancers

Transcriptional activity from 4,599,881 fragments that were 400–500 bp in size, spanning the *Drosophila melanogaster* genome at 17.39X coverage, was analyzed by 100 bp bins and 31 significant hypoxic enhancer regions ($q$-value $< 0.1$, $p$-value $< 1.55e{-}05$) were identified (Table 1, File S5). These enhancer regions range in size from 100 to 800 bp and confer 2 to 18-fold changes in expression under hypoxia. The discovered enhancers are found throughout the genome and are located proximally to genes up-regulated under hypoxia in our RNAseq experiments. The ten most strongly up-regulated genes all contain a discovered enhancer within 20 kb. 16 of 31 discovered enhancers are located within 20 kb of one of the 90 up-regulated genes. The probability of this positional overlap occurring by chance is $1.43e{-}14$ using an exact binomial test, supporting that the discovered enhancers are linked to endogenous gene expression and implicating their likely targets. Four additional enhancers are proximal to genes previously observed to be up-regulated under hypoxia in *Drosophila* (*Li et al., 2013*).

### Location of hypoxic enhancers

Of the 20 hypoxic enhancer regions proximal (within 20 kb) to hypoxic up-regulated genes, 6 fall in the promoter region of the putative target gene (Fig. 2, Table 1). All six of these are the homologous Hsp70B enhancers. Six enhancers were found in introns of putative target

**Table 1  Properties of discovered hypoxic enhancers.** The 31 hypoxic enhancers identified by our genome-wide screen are shown in order of statistical significance. Column one is the genomic location of the enhancer (dm3). Column two is the *p*-value between hypoxic and normoxic counts as calculated by the negative binomial test with column three showing the Benjamini-Hochberg adjusted *p*-value. Column four is the fold change of transcriptional activity due to the enhancer in hypoxic versus normoxic conditions. Column five shows endogenous genes within 20kb that were significantly up-regulated under hypoxia in the same RNA extracts used to calculate enhancer activity. The rank of the gene's hypoxic induction is shown in parentheses and genes marked with an asterisk were observed to be up-regulated under hypoxia in *Drosophila* by *Li et al. (2013)*. Column six indicates the relative position of the enhancer to the proximal hypoxic up-regulated gene. Column seven shows binding sites for stress-related transcription factors found in the enhancer.

| Enhancer locus | *P*-value | Adjusted *P*-value | Fold change | Hypoxic gene(s) within 20 Kb | Relative position to hypoxic gene(s) | Stress TF binding sites |
|---|---|---|---|---|---|---|
| 3R:8303000..8303500 | 7.79e–22 | 4.63e–16 | 5.08 | Hsp70B genes (1–4) | Intergenic | Hsf, Hif-1, Foxo |
| 3L:6256700..6257200 | 1.83e–16 | 2.72e–11 | 5.95 | impl3 (9) | Upstream | NF-kB |
| 3R:8331100..8331800 | 1.59e–16 | 2.72e–11 | 4.49 | Hsp70Bb (2) | Promoter Proximal | Hsf, Hif-1, Foxo |
| 3R:8293200..8293900 | 2.96e–16 | 3.51e–11 | 3.83 | Hsp70Ba (4) | Promoter Proximal | Hsf, Hif-1, Foxo |
| 3R:8334400..8335000 | 1.18e–15 | 1.01e–10 | 4.45 | Hsp70Bc (1) | Promoter Proximal | Hsf, Hif-1, Foxo |
| 2L:8001300..8001800 | 2.64e–15 | 1.74e–10 | 6.44 | Wwox (15) | Intronic | Hif-1 |
| 3R:8327800..8328500 | 8.89e–13 | 2.40e–08 | 3.70 | Hsp70Bbb (3) | Promoter Proximal | Hsf, Hif-1, Foxo |
| 2L:20082900..20083500 | 1.08e–12 | 2.79e–08 | 6.35 | Fok (11) | Intronic | Foxo, Hif-1 |
| 3L:8685300..8685800 | 1.07e–10 | 2.18e–06 | 3.79 | Hairy (45) | Downstream | Hsf, Hif-1, Foxo |
| 3L:7797800..7798600 | 1.77e–10 | 3.38e–06 | 3.07 | CG32369 (23) | Intronic | Hif-1 |
| 3L:9385200..9385800 | 2.14e–09 | 3.62e–05 | 3.71 | Hsp22,23,26,27 (7,8,10,14) | Neighboring Intron | Not Detected |
| X:17071000..17071300 | 8.77e–09 | 1.24e–04 | 4.99 | Not Detected | Not Detected | Not Detected |
| X:9767000..9767500 | 1.27e–08 | 1.76e–04 | 3.65 | CG32695* | ORF | Not Detected |
| 2L:2887100..2887600 | 1.32e–08 | 1.79e–04 | 5.82 | Not Detected | Not Detected | Hif-1 |
| 3L:11234100..11234900 | 6.03e–07 | 6.63e–03 | 2.68 | Scylla (19) | Upstream | Foxo |
| 3L:3892900..3893100 | 1.55e–06 | 1.59e–02 | 2.75 | Not Detected | Not Detected | Hif-1, NF-kB |
| 2L:5986900..5987500 | 1.82e–06 | 1.81e–02 | 2.16 | ifc* | Intronic | Foxo |
| 3L:9448800..9448900 | 2.09e–06 | 2.03e–02 | 5.39 | MTF-1* | Neighboring Intron | NF-kB, Hif-1 |
| 3R:6800900..6801600 | 2.22e–06 | 2.09e–02 | 13.82 | Not Detected | Not Detected | Hif-1 |
| 3L:11522800..11523300 | 2.66e–06 | 2.35e–02 | 3.04 | Not Detected | Not Detected | NF-kB |
| 3R:4181100..4181600 | 2.66e–06 | 2.35e–02 | 3.87 | Atg13 (51) | Downstream | Foxo, Hif-1 |
| 3R:7781900..7782700 | 2.69e–06 | 2.35e–02 | 4.96 | Hsp70Aa (6) | Promoter Proximal | Hsf |
| 3R:7783900..7784500 | 2.75e–06 | 2.37e–02 | 4.18 | Hsp70Ab (5) | Promoter Proximal | Hsf |
| 3R:21433600..21434000 | 3.30e–06 | 2.72e–02 | 9.03 | Not Detected | Not Detected. | Not Detected |
| X:16559200..16559700 | 4.13e–06 | 3.23e–02 | 6.56 | Not Detected | Not Detected | Foxo |
| 3R:2902300..2902600 | 6.21e–06 | 4.63e–02 | 2.95 | Not Detected | Not Detected | Not Detected |
| 2R:12896000..12896500 | 6.88e–06 | 5.05e–02 | 3.02 | Not Detected | Not Detected | Foxo |
| X:17388000..17388500 | 8.24e–06 | 5.75e–02 | 6.80 | Not Detected | Not Detected. | Hif-1 |
| 3R:14892300..14892800 | 9.76e–06 | 6.44e–02 | 18.01 | Not Detected | Not Detected | Hif-1 |
| 3R:27050000..27050500 | 1.52e–05 | 9.40e–02 | 2.78 | CG12054* | Intronic | Hif-1 |
| 3R:25921500..25922100 | 1.54e–05 | 9.44e–02 | 2.46 | Hif-1 (71) | Intronic | NF-kB, Hif-1 |

**Table 2  ncRNAs proximal to hypoxic enhancerst.** Three of the five enhancers not contained within protein coding transcripts coincide with ncRNAs. Each of these ncRNAs is also up-regulated under hypoxia.

| Enhancer locus | ncRNA | Position of ncRNA relative to enhancer | Hypoxic read counts | Normoxic read counts |
|---|---|---|---|---|
| 3R:8303000..8303500 | CR32865 | overlapping | 66 | 13 |
| 3L:8685300..8685800 | CR44526 | 3 bp upstream | 31 | 14 |
| 3L:6256700..6257200 | CR44522 | 201 bp upstream | 6 | 1 |

**Table 3  *P*-value of stress transcription factor binding site enrichment in discovered enhancer sequences**

| Transcription factor | *P*-value of enrichment |
|---|---|
| HSF | $6.22e{-}12$ |
| Hif-1 | $6.49e{-}06$ |
| Foxo | $1.01e{-}04$ |
| NF-kB | $6.67e{-}04$ |

genes (Table 1). These intronic enhancers may be placed proximal to alternate transcription start sites in order to confer isoform specific up-regulation as seen in the case of Sima, the *Drosophila* HIF-1α homologue (Fig. 3). Two enhancers were found in introns of genes neighbouring the putative target and one was found in the ORF of the putative target. The remaining five were found in intergenic space up or downstream of putative target genes, as seen for the enhancer region 13 kb downstream of the transcriptional regulator hairy (Fig. 4). Interestingly, three of the five intergenic enhancers were located immediately proximal to a ncRNA. All of these ncRNAs were themselves up-regulated under hypoxia (Table 2).

## Transcription factor binding motifs

Identified enhancer regions are enriched for binding sites of stress response transcription factors involved in hypoxia. Transcription factors HSF, HIF-1, FOXO, and NF-kB showed highly significant global enrichment across the enhancer regions (Table 3). Binding sites occurring in each individual enhancer are listed in Table 1. 26 of 31 enhancer regions contain binding motifs for at least one of these transcription factors and many contain binding sites for several. In addition to a pair of HSF binding sites, The Hsp70B promoter proximal enhancers contain binding sites for FOXO and HIF-1 (Fig. 2). The intronic Sima enhancer (Fig. 3) contains a pair of HIF-1 binding sites, possibly allowing autoregulation, and also contains a NF-kB binding site. The enhancer region downstream of hairy contains HSF, FOXO, and HIF-1 binding sites (Fig. 4).

## Overlap with STARR-seq enhancers

Our data correlate strongly with a previous genome-wide empirical assay of Drosophila transcriptional enhancers. STARR-seq (*Arnold et al., 2013*) was used to identify 5,499 enhancers operating in S2 cells under normal conditions. These enhancers were defined as

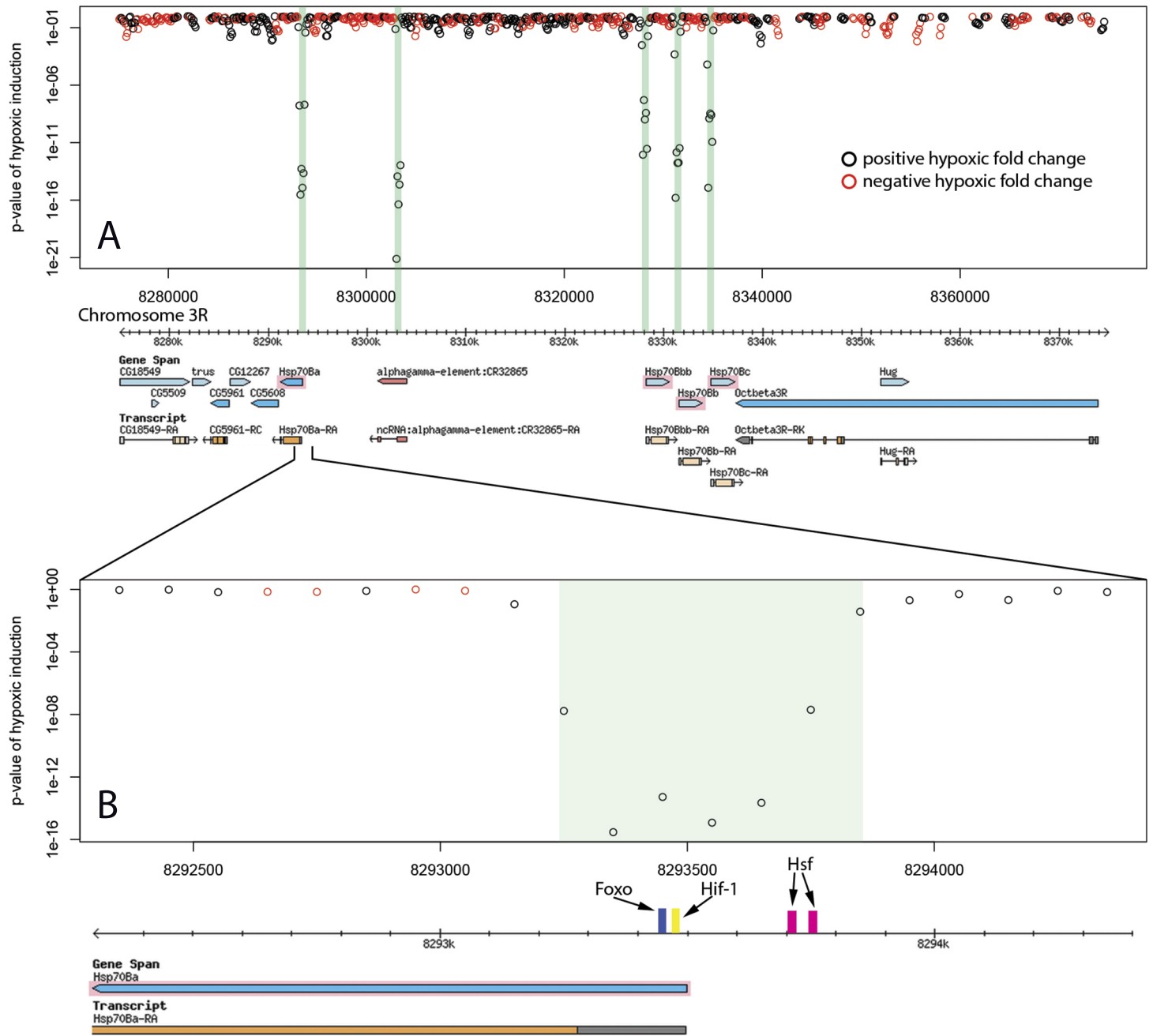

**Figure 2** **Hypoxic enhancer activity by 100 bp bins at the Hsp70B locus.** Each open circle plots the *p*-value of the difference in randomer tag counts mapping to that 100 bp bin between normoxia and hypoxia. Green bars show enhancer regions discovered by our genome-wide screen. (A) The four Hsp70B homologues highlighted in pink are all up-regulated under hypoxia and contain homologous promoter proximal hypoxic enhancer regions. Additionally, a fifth homologous enhancer region lacking an ORF was discovered at the locus. (B) The close up of the Hsp70Ba enhancer region shows the position of multiple stress response transcription factor binding sites.

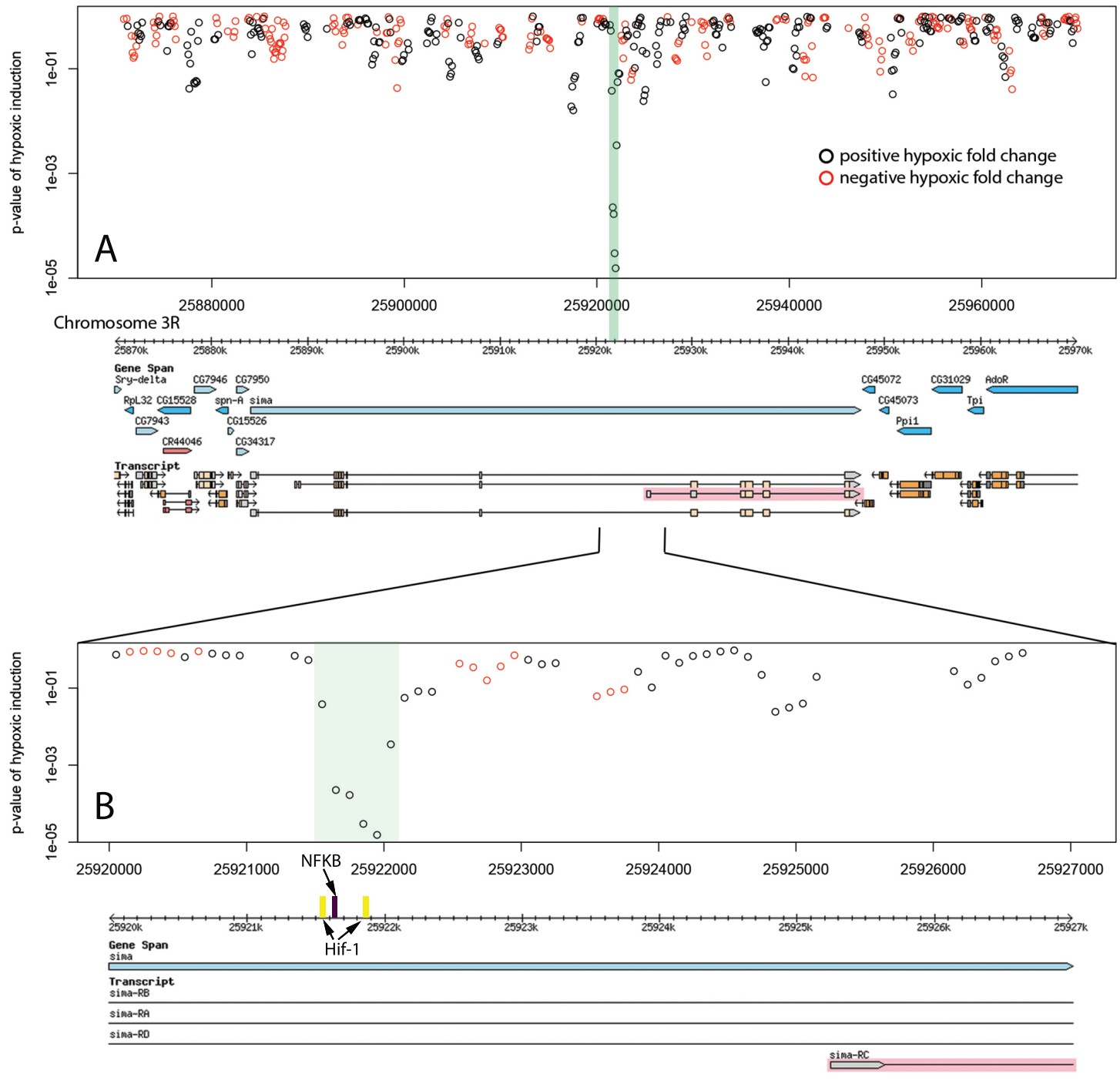

**Figure 3** **Hypoxic enhancer activity by 100 bp bins at the sima (HIF-1 α) locus.** Each open circle plots the *p*-value of the difference in randomer tag counts mapping to that 100 bp bin between normoxia and hypoxia. The green bar shows the enhancer region discovered by our genome-wide screen. (A) HIF-1 is the master hypoxic regulator and is itself regulated transcriptionally under hypoxia. Our RNASeq data shows hypoxia induces up-regulation of the isoform highlighted in pink. We identify an intronic hypoxic enhancer upstream of the transcription start site of this isoform. (B) The close up of the Sima intronic enhancer region shows both HIF-1 and NF-kB binding sites.

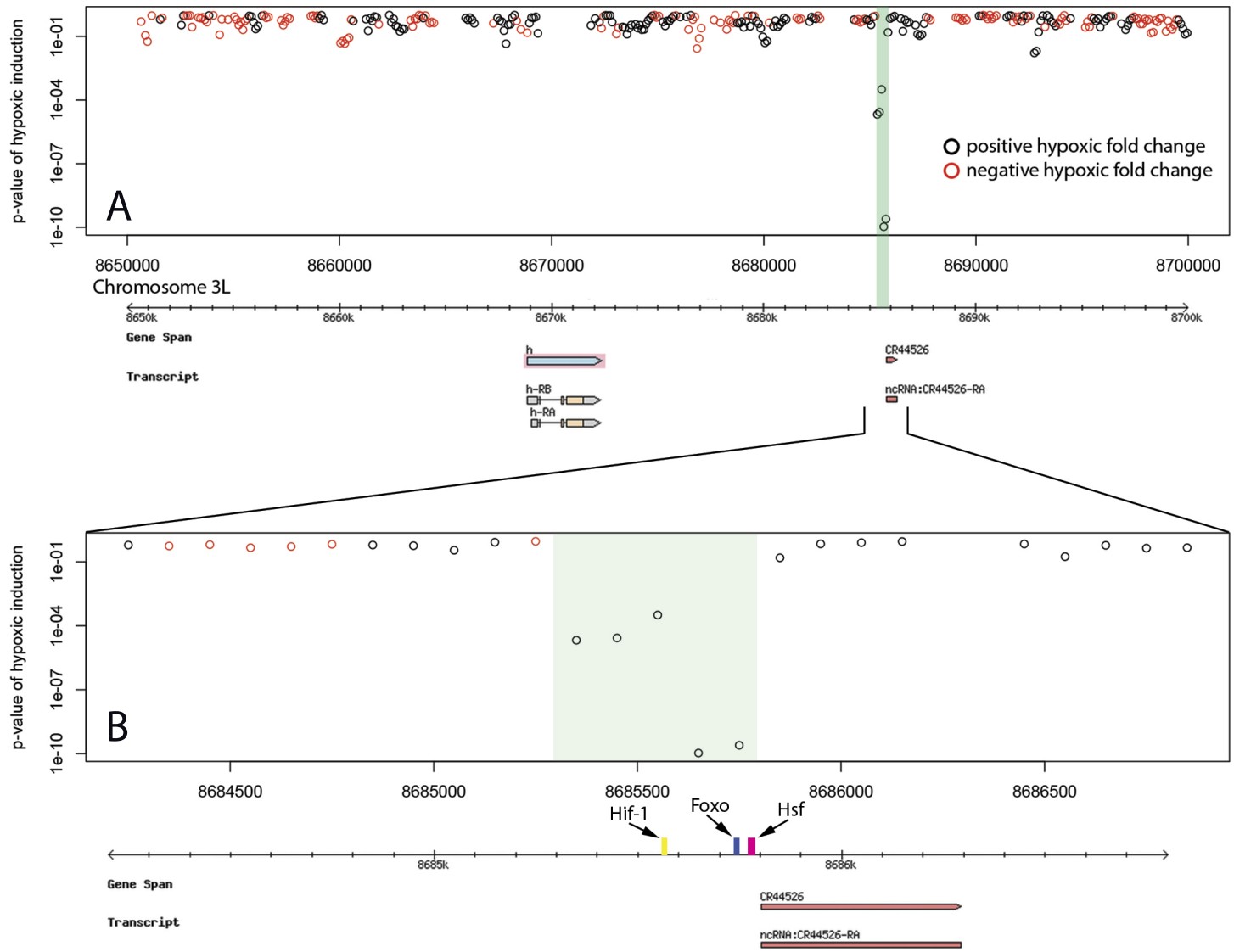

**Figure 4** **Hypoxic enhancer activity by 100 bp bins at the hairy locus.** Each open circle plots the *p*-value of the difference in randomer tag counts mapping to that 100 bp bin between normoxia and hypoxia. The green bar shows the enhancer region discovered by our genome-wide screen. (A) The hairy gene produces a negative transcriptional regulator that is up-regulated during hypoxia. We identify an active hypoxic enhancer 13 kb downstream of hairy. (B) The close up of the hairy downstream enhancer region shows FOXO, HIF-1 and HSF binding sites as well as coincidence with a ncRNA that is also up-regulated under hypoxia.

500 bp intervals surrounding statistically significant peaks in enhancer activity (Adjusted *p*-value < 0.018, *p* < 0.001). In order to generate a similar dataset for comparison, we identified genomic regions showing significant enrichment in normoxic S2 cells. Similar to STARR-seq, we defined enhancers as 500 bp intervals surrounding peaks with an adjusted *p*-value less than 0.018 (unadjusted < 0.00043). This yielded a list of 1,007 baseline S2 enhancers (File S6). A total of 466 of these (46.3%) overlap the enhancers identified using STARR-seq. The probability of a 500 bp fragment overlapping the STARR-seq set by chance is 0.0462. An exact binomial test (463 hits, 1,007 trials, 0.0462 background probability)

generates a *p*-value of 0 for the overlap between our data and STARR-seq. The ratio of overlap is higher when only the most enriched peaks from out dataset are examined. A total of 19 of the 25 most enriched enhancer peaks (76%, *p*-value = 5.66e–21) overlap with the STARR-seq dataset. This high degree of overlap demonstrates a robust ability to have identified active enhancers.

## DISCUSSION

We used a novel parallelized reporter assay to conduct the first genome-wide functional enhancer screen of a cellular response to environmental stress. Our work demonstrates a new method with wide applicability and identifies DNA regulatory sequences conferring hypoxic activity. We identify 31 hypoxic enhancer regions and analyze them with respect to up-regulated hypoxic genes and stress response transcription factors.

RNA-Seq was performed on the same RNA pools used to quantify hypoxic enhancer activity in order to identify putative target genes proximal to identified enhancer regions. Differentially expressed genes identified in our RNA-Seq experiments are corroborated by previous analyses of the *Drosophila* hypoxic response (*Li et al., 2013*; *Liu, Roy & Johnson, 2006*). The majority of enhancer regions were proximal (within 20 kb) to endogenously up-regulated genes, indicating that our enhancer assay identifies active *in vivo* regulatory elements. We identified enhancer regions proximal to previously described hypoxic genes including lactate dehydrogenase (*Bruick & McKnight, 2001*; *Li et al., 2013*), the transcriptional regulator hairy (*Zhou et al., 2008*), the reductase Wwox (*O'Keefe et al., 2011*), and the cell cycle inhibitor scyl (*Scuderi et al., 2006*). Additionally, the Hsp70B promoter proximal enhancers identified in our assay have been previously shown to be active *in vivo* (*Tian, Haney & Feder, 2010*; *Li et al., 2012*). The large positional overlap between up-regulated genes and enhancer regions allowed analysis of the architecture of hypoxic regulation. Interestingly, while some enhancers were found at the promoters of putative target genes, the majority of enhancer regions were found in introns and intergenic space. Enhancers were found in introns of putative target genes as well as introns of neighboring genes (Table 1). Enhancer regions in intergenic space corresponded with known ncRNA loci and in each case the ncRNA was itself up-regulated under hypoxia (Table 2). These findings highlight the unbiased view of the regulatory landscape provided by genome-wide empirical assays and underscore the prevalence of activity outside of promoter regions.

Some of the enhancer regions were not proximal to an identifiable up-regulated gene. These enhancers could act on more distal targets, on proximal targets with expression too low to be detected by our RNA-Seq experiment, or they may have activity in isolation but be attenuated by other elements in their native hypoxic context. Conversely, many up-regulated genes did not have a proximal enhancer identified by our screen. This could be due to a requirement of action from multiple disjunct regulatory modules at the native locus or lack of resolution in our assay. The resolution of our assay was attenuated by the coincidence of randomer tags with multiple genomic regions. Other randomer tag-based approaches test orders of magnitude fewer fragments and hence largely avoid

barcode collision (*Kwasnieski et al., 2012*; *Patwardhan et al., 2012*; *Melnikov et al., 2012*; *Kheradpour et al., 2013*). This problem is circumvented in STARR-Seq (*Arnold et al., 2013*) by confining placement of the potential enhancer to the transcribed region so that it can be assayed directly by RNA sequencing. Future uses of our technique will benefit from further optimization of library synthesis to allow a greater number of randomer tags into the library. Nonetheless, the technique is highly functional in its present state and introduces a simpler and more versatile library synthesis approach. Indeed, our data show a large degree of overlap with STARR-Seq with respect to baseline transcriptional enhancers in *Drosophila* S2 cells. Furthermore, this work presents a large list of empirically identified hypoxia-induced enhancer regions robust to false discovery rate that coincide with the most highly up-regulated hypoxic genes.

The transcription factors HIF-1, HSF, NF-kB , and FOXO regulate hypoxic gene expression and have been shown to exhibit overlapping activity and reciprocal regulation (*Rius et al., 2008*; *Van Uden et al., 2011*; *Scortegagna et al., 2008*; *Hsu, Murphy & Kenyon, 2003*; *Wang, Bohmann & Jasper, 2005*). The enhancer regions identified in this study are highly enriched for their binding site motifs and many display multiple sites allowing signal integration of stress response pathways. We observe an intronic enhancer in Sima which contains both HIF-1 and NF-kB binding sites, suggesting HIF-1 autoregulation and integration of NF-kB signaling at a basal level in the hypoxic response. The enhancer region, while intronic to the full-length Sima transcript isoforms, is upstream of an alternative transcriptional start site that produces a transcript isoform that is up-regulated after hypoxia, whereas the full-length isoforms do not have altered expression after hypoxic stress. This short isoform lacks the bHLH and PAS domains of the full-length isoform, suggesting it neither binds DNA nor heterodimerizes. Interestingly, this hypoxic regulation of a short isoform resembles the hypoxic induction of a short isoform of the HIF-1 regulator *fatiga* (Drosophila HIF-1 Prolyl Hydroxylase) by an intronic HIF-1 enhancer (*Acevedo et al., 2010*).

Our findings reiterate the complexities of the hypoxic response while providing new details. The enhancer regions identified demonstrate regulatory activity distributed throughout non-coding genomic space and underscore the role of intronic enhancers in the hypoxic response. We observe coincidence between enhancer regions and ncRNA activity in agreement with previous evidence showing local transcription to be a general property of active enhancers (*Andersson et al., 2014*). We present a set of sequences capable of driving hypoxia-specific expression and demonstrate a new genome-wide technique for the identification of context-specific enhancers.

## ACKNOWLEDGEMENTS

We thank Paul Etter and Doug Turnbull for advice on Illumina library preparation.

### Funding

This work was supported by the Johnson Laboratory income fund. The funders had no role in study design, data collection and analysis, decision to publish, or preparation of the manuscript.

### Grant Disclosures

The following grant information was disclosed by the authors:
Johnson Laboratory Income Fund.

### Competing Interests

The authors declare there are no competing interests.

### Author Contributions

- Nick Kamps-Hughes conceived and designed the experiments, performed the experiments, analyzed the data, contributed reagents/materials/analysis tools, wrote the paper, prepared figures and/or tables, reviewed drafts of the paper.
- Jessica L. Preston and Melissa A. Randel performed the experiments.
- Eric A. Johnson conceived and designed the experiments, analyzed the data, wrote the paper, prepared figures and/or tables, reviewed drafts of the paper.

### DNA Deposition

The following information was supplied regarding the deposition of DNA sequences:
GenBank:

SRX467593

SRX467591

SRX468157

SRX468694

SRX468097

### Data Availability

The raw data is composed of short read sequences deposited in the NCBI short read sequencing archive. The code used for processing raw data is available in Supplemental Information.

### Supplemental Information

Supplemental information for this article can be found online at http://dx.doi.org/10.7717/peerj.1527#supplemental-information.

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
