# Peer review of "Genome-wide identification of hypoxia-induced enhancer regions"

_PeerJ, doi:10.7717/peerj.1527_

## Round 0.1 · original submission · Minor Revisions

· Academic Editor

Minor Revisions

The study is very well designed and the manuscript is very well-written. Before accepting the manuscript, please answer all the questions raised by reviewers.

·

Basic reporting

Good way of reporting.

Experimental design

Standards are adhered.

Validity of the findings

Standards are adhered.

Additional comments

Authors illustrates a genome-wide method for de novo identification of enhancer regions. They took case of hypoxia-activated enhancers in flies as an example. This study set up a model study for identification of enhancers in other organisms. I give authors full credit for this work. However, I have some points for improvement:

Major points:
1) Abstract: First sentence and last sentence of abstract have same message. I would recommend to develop the needs for this work in the first one or two sentences of the abstract.
Also give a numbers of identified enhances in the abstract.
2) Introduction: Please explain of logic for using hypoxia as an example and why authors have used Drosophila genome

Minor points:
a) Rephrase L200-L223
b) L291: Please "identify" to "have identified"

Reviewer 2 ·

Basic reporting

The manuscript entitled Genome-wide identification of hypoxia-induced enhancer regions represents a very planed and executed study in an area of research that is relevant and of significance.

Authors have reported about the development of a novel method for high through put/ genome wide enhancers. The model activity used for such analysis in the current manuscript i.e. hypoxic transcriptional dynamics makes for a very interesting reading. The experiments performed and data analyses are thorough and scientifically sound.

The overall quality of the manuscript is excellent. It certainly meets the standards of publications of high quality journals.

In its current state, the manuscript is good enough to be recommended for acceptance for publication.

Experimental design

The experimental design is sound and thoroughly represented. The necessary statistical analyses has been performed.

Validity of the findings

The results represented in the present manuscript are relevant and have been validated within the study. Authors have performed complementary experiments for validating the data/ findings. Noticeably, the results have also been cross validated with respect to a previous report that had shown enhance analyses using STAR- seq platform.

Additional comments

Overall the manuscript is well written and describes a very important and interesting development in the research field. I am delighted to recommend the manuscript for acceptance in its current state. Yet authors may think of following suggestions:

1: It would be useful for understanding of general audience to include a graphical representation/ schematics of the entire methodology.

2: Authors should specify what kind of analyses was performed on "R".

3: In the discussion section, authors should give a brief remark about the observed fold change in the activity of identified enhancer (Table 1). It is not evident as to why fold change of enhancer corresponding to HIF -1 is relatively lesser as compared to many other identified enhancers. Also what relevance does this information have with regards to overall signaling pattern of the hypoxia treatment.

4: Authors may represent positive and negative hypoxic fold change on + and - Y axis relative to normoxia. It would make figures 2, 3, 4 much easier to interpret.

5. Authors should indicate whether a 17.39 X coverage is sufficient for genome wide survey of Drosophila genome. Also, indicate how much of coverage would be required for more complex genomes. Such a statement would be beneficial for future users of this newly development platform.

·

Basic reporting

Authors of the paper “Genome-wide identification of hypoxia-induced enhancer region”, used RNAseq data to investigate the enhancer activity for hypoxia at genome-wide scale and reported novel enhancers in drosophila. They have carried out statistical analysis of RNAseq data using DESeq R-package, which uses negative binomial to model “counts of reads”, refers to the number of reads originating from transcripts. In Higher eukaryote reads are frequently ambiguous as to their transcript of origin because of multi-isoform genes and genes families. Hence, transcripts counts to genomic features (enhancer in this case) can be best estimated by negative bionomial which account for both technical and biological variations. Moreover, they have compared their results for the baseline using STARR-seq in their normoxic replicates to confirm their novel enhancer activity in Hypoxia condition. Authors have carried out rigorous and robust analysis to re-confirms their findings.

Experimental design

No comments

Validity of the findings

No comments

Additional comments

Well-written and clear to follow

---

## Round 0.2 · accepted · Accept

· Academic Editor

Accept

The manuscript is now suitable for publication in PeerJ.

·

Basic reporting

OK

Experimental design

OK

Validity of the findings

OK

Additional comments

After these changes, I have no comments.

Reviewer 2 ·

Basic reporting

The revised version of the manuscript is quite appropriate for publication in journal of repute.

Experimental design

The experimental design used in this study is quite appropriate and sufficient.

Validity of the findings

The finding of this report have been cross verified by the authors by comparison with previous reports. The findings are valid and suitable for a high quality publication.

Additional comments

I would like to thank the authors for taking care of comments/ suggestions recommended on the previous version of the manuscript. The updated version of the manuscript along with the rebuttal document clearly indicates the efforts devoted for improvement of the manuscript that was already extremely well prepared.

I would like to congratulate you all for such a neat work and great manuscript.